# Field-based tree mortality constraint reduces estimates of model-projected forest carbon sinks

Kailiang Yu [1,2 ✉], Philippe Ciais [1,3], Sonia I. Seneviratne [4], Zhihua Liu[2], Han Y. H. Chen [5], Jonathan Barichivich[1,6], Craig D. Allen [7], Hui Yang[1,8], Yuanyuan Huang[1,9] & Ashley P. Ballantyne[1,2]

Considerable uncertainty and debate exist in projecting the future capacity of forests to sequester atmospheric $CO_2$. Here we estimate spatially explicit patterns of biomass loss by tree mortality (LOSS) from largely unmanaged forest plots to constrain projected (2015–2099) net primary productivity (NPP), heterotrophic respiration (HR) and net carbon sink in six dynamic global vegetation models (DGVMs) across continents. This approach relies on a strong relationship among LOSS, NPP, and HR at continental or biome scales. The DGVMs overestimated historical LOSS, particularly in tropical regions and eastern North America by as much as 5 Mg ha$^{-1}$ y$^{-1}$. The modeled spread of DGVM-projected NPP and HR uncertainties was substantially reduced in tropical regions after incorporating the field-based mortality constraint. The observation-constrained models show a decrease in the tropical forest carbon sink by the end of the century, particularly across South America (from 2 to 1.4 PgC y$^{-1}$), and an increase in the sink in North America (from 0.8 to 1.1 PgC y$^{-1}$). These results highlight the feasibility of using forest demographic data to empirically constrain forest carbon sink projections and the potential overestimation of projected tropical forest carbon sinks.

[1] Le Laboratoire des Sciences du Climat et de l'Environnement, IPSL-LSCECEA/CNRS/UVSQ Saclay, Gif-sur-Yvette, France. [2] Department of Ecosystem and Conservation Sciences, University of Montana, Missoula, USA. [3] The Cyprus Institute, Nicosia, Cyprus. [4] Institute for Atmospheric and Climate Science, ETH Zürich, Zürich, Switzerland. [5] Faculty of Natural Resources Management, Lakehead University, Thunder Bay, ON, Canada. [6] Instituto de Geografía, Pontificia Universidad Católica de Valparaíso, Valparaíso, Chile. [7] Department of Geography and Environmental Studies, University of New Mexico, Albuquerque, NM, USA. [8] Max Planck Institute for Biogeochemistry, Jena, Germany. [9] CSIRO Oceans and Atmosphere, Aspendale, Australia. ✉email: kai86liang@gmail.com

F orests are major drivers of biophysical land-atmosphere feedbacks, the global carbon and water cycles, and thus overall planetary climate regulation[1]—yet large uncertainties and substantial debate persist regarding ongoing and potential changes in the capacity of forests carbon sinks across continents or biomes (i.e., tropical vs boreal) under anticipated future climates[2,3]. It is often thought that increased net primary productivity (NPP) across spatial resource gradients or under elevated $CO_2$ will increase carbon sinks[4,5]. The loss of ecosystem carbon to the atmosphere through ecosystem heterotrophic respiration (HR), however, has been observed to increase worldwide[6] and could lead to a trade-off or even outweigh the positive effects of NPP in some regions[7], thus potentially amplifying warming through increased rates of ecosystem carbon loss to the atmosphere[8,9]. Reducing uncertainties about the future forest carbon sink thus requires an integrated understanding of NPP and HR across space or time.

Recent studies have suggested that faster forest growth (NPP) leads to higher tree mortality at local and regional scales, particularly in tropical forests[10,11]. This has been recently demonstrated across spatial scales in boreal forests using tree-ring datasets[12] and is consistent with ESM projections across forest biomes[13]. While the fraction of biomass subjected to tree mortality is often treated as a proportion (constant or varying) of standing stocks in ESM simulations[13], this simple model representation of mortality and growth is commonly not observed in forests responding to global change. Ongoing climate change can result in disproportionate mortality relative to growth, from direct physiological mortality from more extreme drought and heat events, insect outbreaks, windthrow, lightning, and wildfire[14,15]. In contrast, growth may be exceeding mortality in regrowing secondary forests[16] or due to increased atmospheric $CO_2$ reducing drought stress[17,18] and/or nitrogen deposition enhancing growth[19]. At long-term and broad spatial scales, however, NPP, biomass loss from tree mortality (hereafter LOSS) and HR could be coupled with their strong positive relationships because of the linkage of aboveground and belowground processes[20,21]. The instantaneous rates of NPP, LOSS and HR, however, could be decoupled over short term or local scales[22,23] and these couplings could be altered with changed availability and/or usage efficiency of resources (i.e., water and nutrients) under global change[18,24]. Thus, data on LOSS which can be directly measured in ground-based forest plots[13] and linked or correlated with NPP and HR, may provide an unique constraint on the forest carbon sink in a future changing climate.

Uncertainty in projections of forest carbon cycling could be reduced through an emergent constraint (EC) approach by identifying heuristic relationships between a multi-model ensemble and an observational estimate. The essence of such an EC approach is to examine the statistical relationships between historical and projected variables of interest in a multi-model ensemble, whereby the historical observations are used to reduce the uncertainty of model projections[25]. This empirical EC approach is complementary to the bottom-up approach in which data or process optimization (sometimes through data assimilation) is applied to improve model projections[26]. This EC approach has been used to constrain the projected land carbon storage[27], or photosynthesis/GPP[28–31], with the assumption that the processes driving the long-term response are also driving the historical (short-term) patterns[27,28]. Interpreting the results of EC requires caution in the confirmation of the verified mechanisms[25,32] and addressing the mismatch of spatial and temporal scales between data and models[25,30]. Earlier studies usually used atmospheric variables (i.e., $CO_2$ concentrations) to constrain carbon storage and photosynthesis at regional to global scales[27,28]. Using spatially explicit observational products (i.e., GPP, evapotranspiration and

leaf area index), studies have emergently constrained the future terrestrial carbon cycling projections at grid scale and then spatially aggregated to broader spatial scales[29,33]. Recent studies highlight the potential of integrating emergent constraints at lumped broad spatial scales and machine learning to generate a spatially explicit constraint on projected gross primary production (GPP) by accounting for the non-linear relationships between GPP and environmental drivers[34]. However, to date it remains unclear how ground-based datasets, such as forest plot observations, could better constrain the projected forest carbon sink at broad spatial scales in a future climate. Because observational uncertainty has more influence than model ensemble uncertainty in EC[25], a LOSS constraint based on forest plot data is expected to reduce the uncertainties in projections of future forest carbon sinks and associated feedbacks to climate.

Here we generate spatially explicit patterns of LOSS from long-term (1951 to 2018) forest plot data (n = 2676; Supplementary Fig. 1) to constrain projected (2015–2099) NPP, HR and net ecosystem exchange (NEE = NPP – HR) in six DGVMs across continents - North America, South America, Africa and Asia & Australia. The data are from largely unmanaged forest plots, because management implies distinct LOSS patterns, and management/forestry is not incorporated in models. Our approach is motivated by the positive heuristic relationships (Supplementary Figs. 2 and 3) between historical LOSS and projected NPP and HR carbon fluxes in DGVMs, in line with the known pattern of faster growth and higher mortality[10–13] and the couplings of growth, mortality and respiration at long-term and broad spatial scales[20,21]. We used LOSS also because: (1) it can be directly measured in forest inventory datasets[13] with high accuracy relative to remote sensing; (2) LOSS remains less studied relative to NPP and LOSS is unrealistically represented (i.e., as a proportion of NPP) in DGVMs[13]. Thus, more work with LOSS data products is urgently needed in scientific communities; and (3) the accuracy of constraining the projected carbon cycle in DGVMs largely depends on the observational uncertainty[25]. We first used a random forest to upscale spatial variations of LOSS with 57 environmental variables (see Supplemental Data 1) to generate a spatially explicit map of LOSS at 0.25-degree resolution across continents (see Methods). Second, we compared the observed spatial patterns of LOSS with the patterns of DGVM[35] in which tree mortality was explicitly reported (see Methods). Finally, two complementary approaches—a conventional emergent constraint and a machine learning constraint—were used to constrain the projected NPP and HR at continental scale in DGVMs (see Methods, Supplementary Fig. 4). The conventional emergent constraint (EC) approach was applied by identifying a statistical (linear) relationship between historical LOSS, aggregated or averaged at local forest-plot scale (using original forest plot data) or continental scale (using upscaled LOSS maps), and projected NPP and HR, summed at continental scale across the DGVM ensemble. In this sense, each model was treated as a sample to fit the heuristic (linear) relationship between LOSS and projected NPP and HR and the observational LOSS (mean ± sd) was then used to impose the constraint on projected NPP and HR (see Supplementary Fig. 4 for more details). The mismatch of spatial scale in the conventional EC approach highlights the need for caution in interpreting results[25]. By comparison, the machine learning (ML) approach was used to examine the non-linear relationships by training a ML (random forest) model between historical simulated LOSS and projected NPP and HR for each DGVM at grid scale. This ML approach thus allowed for inclusion of all gridded patterns of LOSS and a spatially explicit constraint of projected NPP and HR, while we note that the couplings of LOSS, NPP and HR could be weak at local or pixel scales.

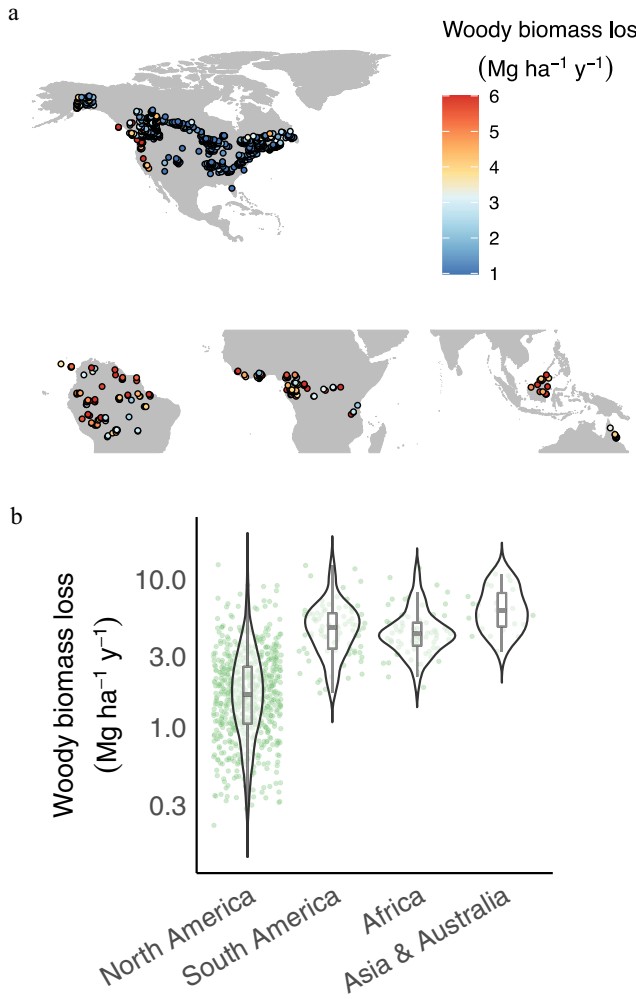

**Fig. 1 Map of sample locations and biomass loss to mortality (LOSS) data. a** Sampling sites. A total of 2676 samples were collected and aggregated into 814 grids at 0.25 degree that were used for geospatial modeling. **b** The median and interquartile range of LOSS across continents —North America, South America, Africa, and Asia & Australia.

## Results and discussion

**Biogeographic pattern of LOSS.** The original forest plot data aggregated at 0.25 degree show large spatial variations (Fig. 1a) across the continents, with the greatest LOSS in Asia & Australia (mean ± 1 SE; $6.5 ± 0.5$ Mg ha$^{-1}$ y$^{-1}$) > South America ($4.9 ± 0.2$ Mg ha$^{-1}$ y$^{-1}$) and Africa ($4.6 ± 0.2$ Mg ha$^{-1}$ y$^{-1}$) > North America ($2.3 ± 0.1$ Mg ha$^{-1}$ y$^{-1}$ in boreal and $2 ± 0.1$ Mg ha$^{-1}$ y$^{-1}$ in temperate)[36] (Fig. 1b; Supplementary Fig. 5a). This pattern was robust to bootstrapping (1000 iterations) to randomly select 90% of plots for estimating the probability distribution of the mean continental values (Supplementary Fig. 5b). The upscaled gridded LOSS maps generated by our random forest algorithm (see Methods) over the spatial domain of our full datasets shows hotspots of high LOSS in Southern Asia & Australia (> 6 Mg ha$^{-1}$ y$^{-1}$), Northwestern South America (Amazon) (> 5 Mg ha$^{-1}$ y$^{-1}$), and the western coast of North America (>3 Mg ha$^{-1}$ y$^{-1}$)[10,36–38] (Supplementary Fig. 6a). These patterns were robust to two bootstrapping approaches – based on the sampled biomes of each point feature and also randomly sampling 90% data with replacement (see Methods) (Fig. 2a; Supplementary Fig. 6b). The uncertainty (coefficient of variance - CV; %mean) was generally low (<10%) across continents, with the exception of temperate forests in North America (CV > 10%), despite the larger sample size ($n > 500$ at 0.25 degree)

(Fig. 2b; Supplementary Fig. 6c), likely because of potential effects of forest recovery or regrowth following past disturbance[16] as well as the small plot size (i.e., 0.067 ha in each plot)[39].

**Drivers of LOSS.** Mean annual temperature (MAT), aridity index (the ratio of precipitation to potential evapotranspiration), and precipitation seasonality were identified as the dominant predictors of LOSS across continents (Supplementary Fig. 7a), with positive relationships with LOSS (Fig. 3a)[10,36]. In contrast to local-scale studies[40,41], wood density, forest stand density, and soil conditions were poor predictors of LOSS when all data were used. These relationships were largely driven by the spatial pattern of LOSS and climate gradients, whereby LOSS and MAT, aridity index, and precipitation seasonality were high in tropical forests (Supplementary Fig. 8). This motivated us to examine the drivers of LOSS in tropical vs non-tropical biomes (Supplementary Fig. 7b, c; Fig. 3b–d). With a smaller gradient in climate within wet tropical forests, soil properties such as nutrient content and cation exchange capacity (CEC) were significant predictors of LOSS (Supplementary Fig. 7b; Fig. 3b)[42]. In wet tropical forests, the relationships between soil nutrient content and CEC and LOSS were positive (Fig. 3b) and thus appeared to support the pattern of higher mortality in more productive tropical forests growing over nutrient rich soils[42,43]. In non-tropical regions, basal area or a competition index based on the degree of crowding within stocked areas[44] (see Methods) were the dominant predictors of LOSS, especially in extra-tropical North America (Supplementary Fig. 7c; Fig. 3c, d). This result highlights the role of stand competition in driving the spatial patterns of LOSS[44,45]. This pattern also supports the existence of a spatial tradeoff between faster growth and higher mortality because of resource limitations or younger death, whereby competition plays the fundamental role[13,45]. In contrast to other studies[15,46], forest age (available in boreal and temperate forests in North America) was not a good predictor of LOSS (Supplementary Fig. 9), likely because of our focus on mature and old-growth forests (i.e., age > 80 and 100 years in boreal and temperate forests, respectively).

**Data and model comparisons.** We then compared the observed patterns of LOSS with those simulated by six state-of-the-art DGVMs in which tree mortality and LOSS were explicitly simulated[35]. The results show divergent predictions of LOSS among DGVMs with four models (ORCHIDEE, JULES, LPJmL, and SEIB-DGVM) overestimating LOSS compared to our observation-based estimate, particularly in tropical forests and temperate eastern North America, while CABLE-POP and LPJ-GUESS underestimate LOSS across continents (Supplementary Fig. 10). This led to a model ensemble mean overestimation of LOSS, particularly in tropical forests (historical ΔLOSS > 5 Mg ha$^{-1}$ y$^{-1}$, where Δ is the model minus observed value) and eastern North America (ΔLOSS > 4 Mg ha$^{-1}$ y$^{-1}$) (Fig. 2c, e), while the spread of LOSS prediction between models was greatest (CV > 130%) in western boreal forests in North America (Fig. 2d).

**Conventional emergent constraint.** We first used the conventional emergent constraint approach[27] to constrain the projected (2015–2099) NPP and HR across continents. This approach was conducted by building the statistic (linear) relationship between the historical LOSS averaged at forest-plot scale (derived from original plot data of LOSS) or continental scale (derived from the map of LOSS) and projected NPP and HR summed across continents (see Methods and Supplementary Fig. 4 for details). We found that the emergent constraint approach worked well in North America, where the relationship between historical LOSS and projected NPP and HR was significant (the scenario of using

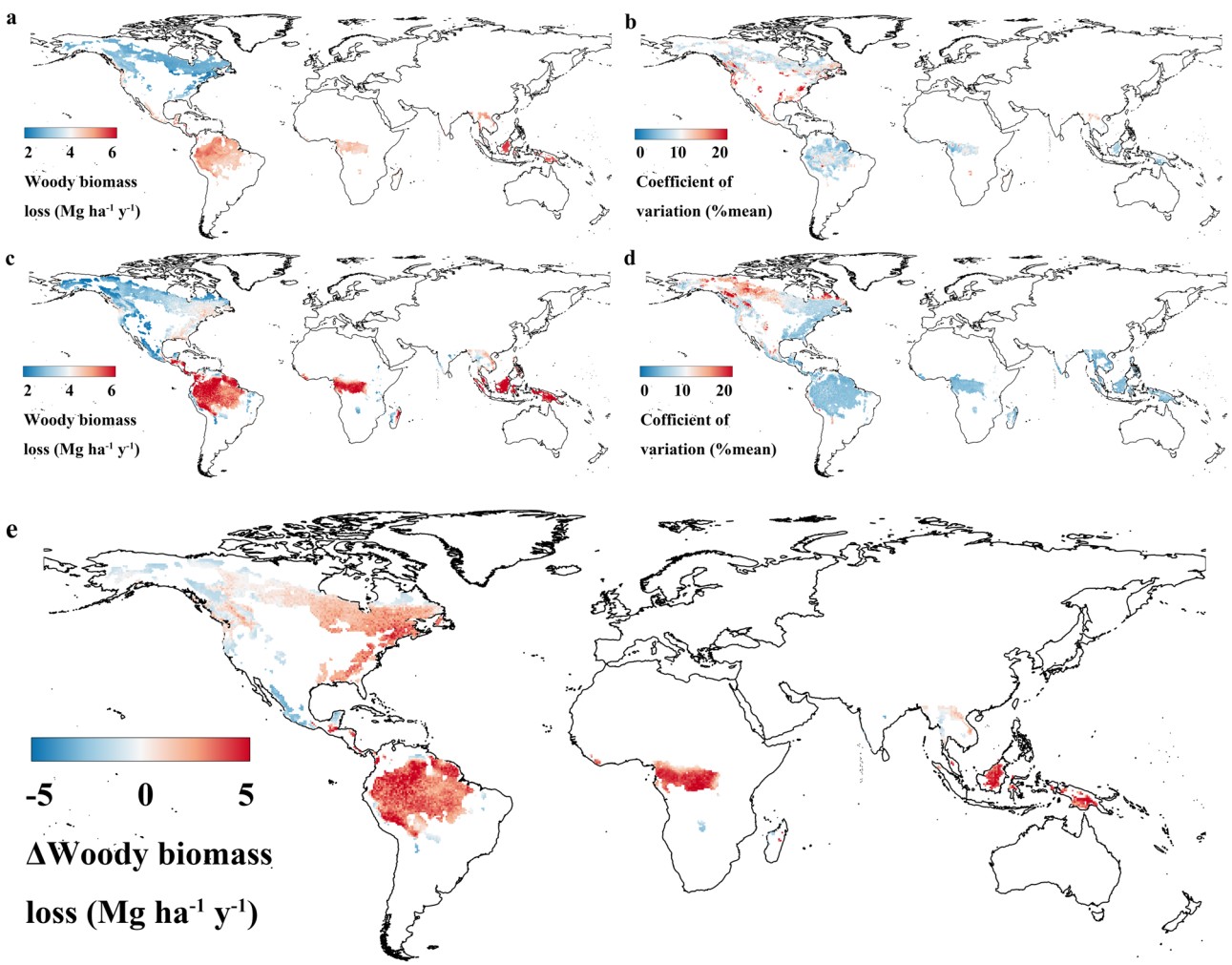

**Fig. 2 Map of biomass loss to mortality (LOSS) and its uncertainty across continents. a**, **b** Ensemble mean of LOSS **a** and its uncertainty (coefficient of variation, **b** across continents at 0.25 degree derived from forest plot data using the bootstrapped (10 iterations) approach by randomly sampling 90% plots with replacement. **c**, **d** Ensemble mean of LOSS **c** and its uncertainty (coefficient of variation, **d** across continents at 0.5 degree derived from six dynamic vegetation models (DGVMs, ORCHIDEE, CABLE-POP, JULES, LPJ-GUESS, LPJmL, and SEIB-DGVM). Coefficient of variation was quantified as the standard deviation divided by the mean predicted value as a measure of prediction accuracy. **e** The difference of LOSS between ensemble mean of DGVMs and ensemble mean of LOSS derived from forest plots data across continents at 0.5 degree, quantified as difference between **c** and **a**, whereby LOSS in Fig. 2a is resampled at 0.5 degree.

original plot data of LOSS: $R^2 = 0.68$ and $P = 0.04$ for grid-level NPP; $R^2 = 0.97$ and $P = 0.0001$ for grid-level HR; the scenario of using map of LOSS at continent scale: $R^2 = 0.7$ and $P = 0.04$ for grid-level NPP; $R^2 = 0.95$ and $P = 0.0008$ for grid-level HR) (Supplementary Fig. 11a; Supplementary Fig. 12a). This emergent constraint approach was less effective, however, for other continents, where tropical forests are predominant (all $P > 0.05$; Supplementary Fig. 11b, c, d; Supplementary Fig. 12b, c, d). These results suggest a weak linear relationships when observations are lumped or averaged at broad continental scales for tropical continents, thus highlighting the importance of spatial scale and non-linear relationships in emergent constraint[25]. We interpret the result that this LOSS emergent constraint works better in North America than in the tropical forests, by a better representation of forest plot distribution and couplings of LOSS and NPP and HR across space in North America.

**Machine learning constraint**. To overcome this limitation, we trained a machine learning algorithm[34] to reproduce the emerging relationship between historical LOSS and projected NPP and

HR at grid level in each DGVM by incorporating all grid values without or with climate predictors, expressed as $NPP_{pro}$ or $HR_{pro} = f(LOSS_{his})$ or $f(LOSS_{his}, MAT_{pro}, MAP_{pro})$), respectively, where $_{pro}$ refers to projected variables, $_{his}$ refers to historical variables, and MAT and MAP is mean annual temperature precipitation, respectively (see Methods). The results show consistently positive non-linear relationships between $LOSS_{his}$ and $NPP_{pro}$ or $HR_{pro}$ across DGVMs (Supplementary Fig. 3). Our machine learning algorithms can surrogate well the results of process-based models between the historical LOSS and the projected NPP and HR ($R > 0.65$ and $R > 0.9$ in both scenarios without climate effects and with climate effects, respectively; see Methods) (Supplementary Fig. 13). After including the observed $LOSS_{his}$ (derived from LOSS) in the machine learning algorithm, we were able to generate spatially explicit constrained estimates[34] of projected NPP and HR, and then compare them with the scenario without the constraint (Supplementary Fig. 14; Supplementary Fig. 15). These patterns essentially show a lower $NPP_{pro}$ or $HR_{pro}$ in locations of overestimated $LOSS_{his}$ in DGVMs, consistent with the positive relationship between $LOSS_{his}$ and $NPP_{pro}$ or $HR_{pro}$ (Supplementary Fig. 3).

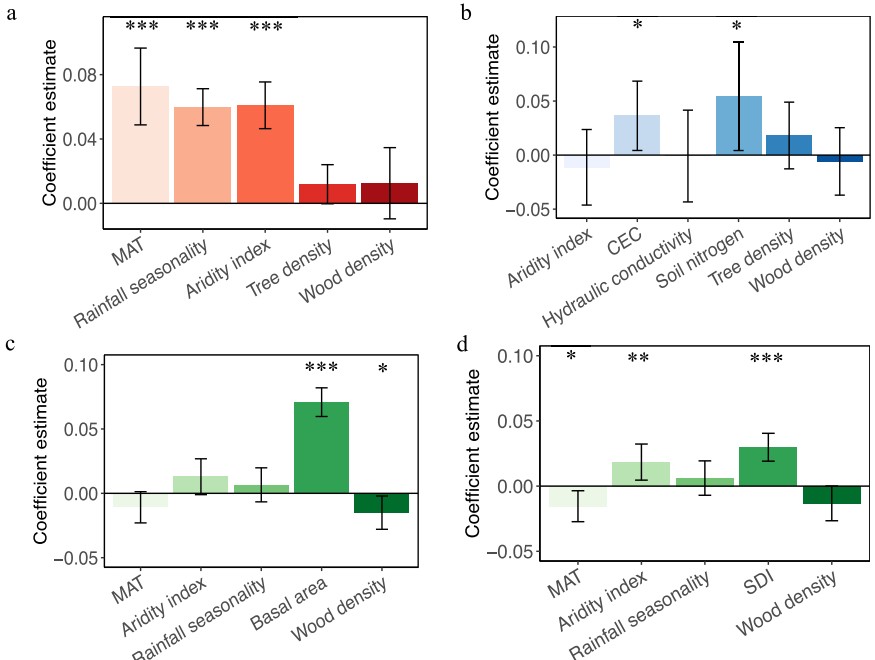

**Fig. 3 Standardized response coefficients (mean ± 95% CIs) between LOSS and dominant environmental drivers.** The scales analyzed were at continents **a**, tropical regions **b** vs non-tropical regions **c**, **d**. The response coefficients were quantified by linear mixed model which account for each plot as a random effect. Panels c and d used basal area and stand density index (SDI) as competition index, respectively. SDI was defined as the degree of crowding within stocked areas and quantified as a function of tree density and the quadratic mean diameter in centimeters. Basal area is strongly correlated with total biomass and higher LOSS in higher basal area may be merely because of its correlations. Thus, we used another competition metrics – SDI to further confirm the role of competition in LOSS. The error bars denote the 95% confidence interval. *$P < 0.05$; **$P < 0.01$; ***$P < 0.001$.

Our results show that most DGVMs overestimate tree mortality, particularly in tropical regions (Fig. 2c, e). Thus, if modeled mortality is over-estimated, we expect that NPP is over-estimated as well. Ultimately, we used a bootstrap approach to generate 100 maps of mean value of LOSS with its distribution following the values of the average and 2 times of standard deviation of LOSS maps as a conservative constraint (see Methods). Then the 100 maps of mean value of LOSS were used to constrain the projected NPP or HR as ensemble means in our ML constraint and the uncertainty of the constraint was assessed. Our bootstrapping constraint approach by LOSS reduces this common bias of models and decreases projected NPP down to 7.9, 2.3, 2 Pg C y$^{-1}$ in South America, Africa and Asia & Australia, compared to original NPP values of 9, 2.4, 2.3 Pg C y$^{-1}$ (Fig. 4a). The reason for this is that NPP or growth is strongly positively correlated with LOSS across space in both inventory data and DGVMs (Supplementary Figs. 2 and 3; Supplementary Fig. 16). The constant mortality parameter used in most models may be too large if modelers have tuned this parameter to obtain reasonable biomass stocks, thus compensating for an over-estimate of NPP in absence of modeled competition between individuals and nutrients (e.g. phosphorus) limitations in tropical forests[13]. Likewise, HR$_{pro}$ showed similar patterns with NPP$_{pro}$ because of coupling of HR and NPP and LOSS at broad spatial and long term scales[20,21], despite the likely decoupling of the instantaneous rate of HR and NPP and LOSS at local and short-term scales[22,23]. Thus, we also constrained a decrease in projected grid-level HR with values of 6.5, 1.9, 1.7 Pg C y$^{-1}$ in South America, Africa and Asia & Australia compared to 7, 1.9, 1.8 Pg C y$^{-1}$ in the original model ensemble (Fig. 4b). Taken together, our results constrain a weaker future tropical forest carbon sink from observation-based LOSS estimates down to 1.4, 0.4, 0.3 Pg C y$^{-1}$ in South America, Africa and Asia & Australia as compared to 2,

0.5, 0.5 Pg C y$^{-1}$ in the original models. The projected sink is reduced in particular over the Amazon basin, while North America showed an enhanced future carbon sink (1.1 and 0.8 Pg C y$^{-1}$ after and before constraint, respectively). The constraint by the machine learning approach significantly reduced the model spread in grid-level NPP$_{pro}$ and HR$_{pro}$ generally in tropical regions and particularly in South America (Fig. 4; Table 1). This was in contrast to the case of constraint at the whole North America scale (Fig. 4; Table 1), presumably because of spatial trade-off or compensation from regions of mortality over-estimation (i.e., eastern North America—temperate zones) vs underestimation (i.e., boreal zones). To this end, we further divided the whole North America into temperate and boreal forests and found the significant effects of the ML constraint (Supplementary Fig. 17). These results highlight the importance of spatial scale in the ML constraint approach. We thus recommend accounting for the role of spatial trade-off in our ML constraint approach or using our ML constraint approach at broad spatial scales whereby the effect of spatial trade-off is minimal. We also caution that the bootstrapping (100 times) approach used in our ML constraint increases the sample size and could have increased the significant difference with and without LOSS constraint. Overall, the uncertainty of the ML constraint was low in the bootstrapping approach (Supplementary Fig. 18).

Our results were robust to the inclusion of projected climate (temperature and precipitation) across space in the machine learning algorithm (Supplementary Fig. 19; Supplementary Table 1), while we note that our approach does not account for effects of atmospheric CO$_2$ concentrations[27,28,30]. Indeed, our study focuses on the carbon flux—NPP and HR and carbon sink averaged over the long-term projected future (2015–2099) in mature forests across continents. The mechanistic basis under-lying this approach is the observed pattern of faster growth and

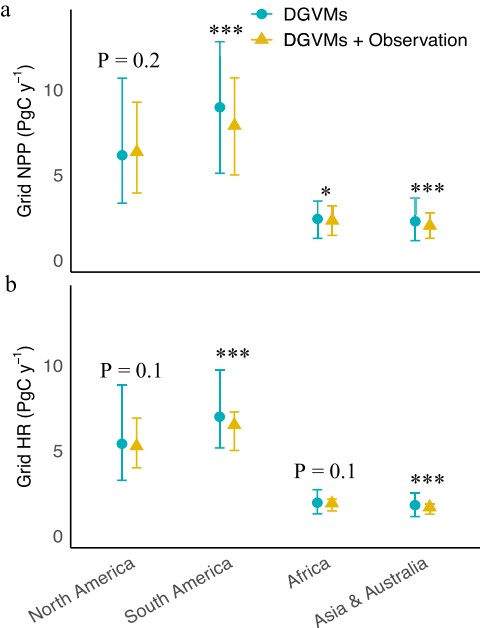

**Fig. 4 Projected grid-level NPP and grid heterotrophic respiration (HR) across continents. a, b** Projected (2015–2099) grid-level NPP **a** and grid-level HR **b** across continents quantified by six dynamic vegetation models—DGVMs (ORCHIDEE, CABLE-POP, JULES, LPJ-GUESS, LPJmL, and SEIB-DGVM). The y axes are the minimum, mean, and maximum values in six DGVMs. 'DGVMs' refers to the scenario before constraint and 'DGVMs + Observation' refers to the scenario after constraint without climate predictors. The constraint was achieved by using the observational maps ($n = 100$; through a bootstrapping approach; see Methods for details) of LOSS derived from forest plots data to feed into the trained ML (random forest) model. Reported are ensemble means of constraint. The constraint effect was significant when North America were divided into temperate and boreal forests (see results of Supplementary Fig. 17). *$P < 0.05$; **$P < 0.01$; ***$P < 0.001$.

**Table 1 Modeled projected (period 2015–2099, units in Pg C y$^{-1}$) grid-level NPP and grid-level HR before and after constrain across the continent in six DGVMs.**

| Continent | Before constraint | | After constraint | |
|---|---|---|---|---|
| **NPP** | **Mean** | **SD** | **Mean** | **SD** |
| North America | 6.1 | 2.9 | 6.4 | 2.1 |
| South America | 9.0 | 3.0 | 7.9*** | 2.1 |
| Africa | 2.4 | 0.8 | 2.3* | 0.6 |
| Asia & Australia | 2.3 | 0.9 | 2.0*** | 0.5 |
| **HR** | | | | |
| North America | 5.4 | 2.0 | 5.3 | 0.9 |
| South America | 7.0 | 1.7 | 6.5*** | 0.7 |
| Africa | 1.9 | 0.5 | 1.9 | 0.2 |
| Asia & Australia | 1.8 | 0.5 | 1.7*** | 0.2 |

Note: (1) the scenario is without accounting for the effects of projected climate—precipitation and temperature;
(2) a bootstrapping (100 times) approach was used for constraint of projected NPP or HR (see Methods) and ensemble means were reported;
(3) the ANOVA test was used to explicitly evaluate whether the difference before and after constrain of projected NPP or HR was significant. * for $P < 0.05$; ** for $P < 0.01$; *** for $P < 0.001$.

higher mortality and thus an emerging coupling of growth, mortality, and respiration averaged over long-term and broad spatial scales, which have been demonstrated in forest inventory datasets[11,47], tree rings[12], eddy flux towers[20] and Earth System

models[13] and DGVMs here. Our approach assumes that this emergent coupling holds both in the short-term historical period and will hold in the future, thus allowing us to use observed historical LOSS to constrain the future projected NPP and HR carbon fluxes.

**Conclusions and implications**. Models typically have a simple representation of tree mortality processes, as a fraction of standing stock[13], but increasingly more detailed global datasets of tree biomass and demography will also make it possible to test more realistic simulations of competition for light, water and nutrients in the next generation of vegetation models[35,48]. The finite resources and vegetation carrying capacity govern the tradeoffs between growth and mortality and respiration across space or time. We note, however, that changes in availability and/or usage efficiency of resources (i.e., water and nutrients) could potentially change the tradeoffs or couplings of mortality and growth across space[18,24], thus highlighting the significance of accounting for resource usage efficiency in the model projections. Another non-modeled factor of decoupling is the role of climate-induced disturbances that could strongly increase LOSS and have a delayed positive or negative effect on NPP in the recovery phase.

HR remains even more poorly understood and more simply represented than LOSS and NPP in DGVMs. Thus, the couplings of LOSS with HR could be weaker or more uncertain (subject to changes) relative to the couplings of LOSS and NPP at long term and broad spatial scales. This presumably explains why our ML constraint has a greater influence on NPP than HR found in this study. Our results show the potential of leveraging a machine-learning (ML) approach of constraint and forest demography data to constrain the projected NPP and HR at broad spatial scales. This ML approach accounts for the non-linearity and all pixel values of variables of interest (i.e., LOSS here), which is not necessarily considered using the conventional EC approach. Our results indicate that the projected increase in tropical forest productivity after constraint may not be as large as previously thought relative to predictions of unconstrained models, especially over the Amazon. These reductions in tropical carbon uptake may offset projected increases in boreal and temperate forest productivity, thereby reducing the model estimates of carbon sink potential of global forests.

**Methods**

**Forest plot datasets**. Forest plot data used in this study met the following criteria[13]: (1) all plots had at least three consecutive censuses and long-term (>9 years) observations between the first and last census so that LOSS averaged over all censuses was representative of the historic forest status; (2) plots were natural, unmanaged forest stands that have not been disturbed by fires, harvesting, and other human activities; (3) the plots were largely mature or old-growth forests and were screened by criteria such as forest age or forest gymnosperm fraction (see Supplementary Information for details); (4) the plots were in a quasi-steady state and were screened by criteria that plots with growth more than 3 times of LOSS or LOSS more than 3 times of growth were excluded—such large differences (i.e., more than 3 times) between LOSS and NPP are likely due to disturbances such as fires that are not well-represented yet in DGVMs. Thus this criterion is used to further select natural mature and old-growth forest stands; and (5) the plots with low values of biomass (i.e., <3 Kg m$^{-2}$) were excluded because they are not fully stocked and thus not likely to be mature forests. Ultimately, we compiled a broad-scale ($n = 2676$) and long-term (1951 to 2018) dataset of largely unmanaged forest plots in a quasi-steady state, distributed in Canada, USA, Amazon, Africa and Asia & Australia (Supplementary Fig. 1). When compared with DGVMs, the above-ground woody biomass loss was converted to total woody biomass loss including belowground roots using the root-shoot biomass ratio product[49]. Otherwise, the original aboveground woody biomass loss (i.e., in the constraint or analysis of drivers of LOSS) was used and reported to avoid the additional uncertainty from the root-shoot biomass ratio product. More details for the criterion of plots selected, plot establishment, and measurements are described in Supplementary Methods.

**DGVMs**. Simulated NPP, mortality flux, and HR of six DGVMs (ORCHIDEE, CABLE-POP, JULES, LPJ-GUESS, LPJmL, and SEIB-DGVM) were publicly available from the carbon turnover inter comparison project https://zenodo.org/communities/vegc-turnover-comp/?page=1&size=20[35]. We used these six DGVMs because they explicitly reported tree mortality. To be comparable with forest plot data, we used the components of tree mortality flux without disturbance, such as fires (Supplementary Table 2). The historic mortality flux was averaged over 1961–2014 forced by CRU-NCEP v5 climate data and the projected NPP and HR were averaged over 2015–2099 forced by bias-corrected IPSL-CM5A-LR RCP 8.5 simulated climate data. Because the output of HR is reported at grid scale and it is not able to be decomposed into components of HR contributed by trees vs nontrees, we used grid scale NPP to estimate forest carbon sink, quantified as the difference of grid-level NPP and grid-level HR. To be consistent with forest inventory data, outputs of historical mortality at tree-level were used to compare with observations. All outputs were resampled and analyzed at 0.5 degree.

**Geospatial modeling and environmental drivers**. We used the Random Forest machine learning algorithm[50] (see Supplementary Methods) with the derived 57 environmental covariates including climate, vegetation and soil conditions (Supplementary Data 1) to extrapolate these relationships between LOSS and environmental conditions across continents and generate the first spatially-explicit and quantitative map of LOSS at the continental scale. The 57 environmental covariates were derived based on the georeferenced coordinates of forest plot data ($n = 814$) aggregated at 0.25 degree. 10-fold cross-validation was used to evaluate the strength of prediction and the best model having high coefficient of determination and low standard deviation in the mean cross-validation were used to generate the map of LOSS. The standard error sharply decreased with increasing sample size across all vegetation biomes and the analysis showed that an efficient prediction required a large sample size ($n > 400$) (Supplementary Fig. 20a). Random Forest was able to predict the variation in LOSS with high predictive accuracy ($R^2 = 0.48$ in 10-fold cross-validation; $R^2 = 0.93$ in final model; Supplementary Fig. 20b). Two types of bootstrapping were used to evaluate the uncertainty (standard deviation as a fraction of the mean predicted value) in the map of LOSS. One was based on a stratified bootstrapping (100 iterations) procedure[51], which was the sampled biomes of each point feature (LOSS) with the total number collection of LOSS points to avoid biases. The second bootstrap was based on randomly sampling 90% with replacement (10 iterations) to account for the biases from an unbalanced sample distribution. In both the two types of bootstrapping, ensemble mean and 95% confidence intervals of LOSS were computed by grid. Rasters of tree cover, human footprint index, percentage of annual burn area and managed land cover (see Supplementary Methods) were used as mask to define the natural forest areas across continents.

To examine the environmental controls of LOSS, we chose the top drivers which include climate conditions (mean annual temperature, MAT; aridity index; precipitation seasonality), vegetation properties (tree density, basal area, competition index, wood density and biome type) and soil properties (soil organic carbon, SOC; soil N; soil hydraulic conductivity, Ks; cation exchange capacity; clay content; and pH) (see Supplementary Methods). Competition index was defined as the degree of crowding within stocked areas and quantified as a function of tree density and the quadratic mean diameter in centimeters[44]. These variables were examined to avoid multicollinearity using a matrix of pairwise correlations to remove any variable with high correlations ($R > 0.7$) with other predictor variables[52] and variation inflation factor (VIF < 4). The Random Forest machine learning algorithm was then used to determine the importance of each predictor variable[53]. Mean decrease in accuracy (%IncMSE) were reported and the variables with greater values of %IncMSE are more important in influencing LOSS. To account for each plot as a random effect, we also used a linear mixed model to examine the dominant factors on LOSS across continents and in tropical regions vs non-tropical regions (see Supplementary Methods).

**Constraining projected forest carbon sinks**. Constraint analyses were conducted at a spatial resolution of 0.5 degree, with model outputs and observational LOSS maps resampled at 0.5 degree. We first attempted to use the conventional emergent constraint (EC) approach to constrain the projected (2015–2099) NPP, HR, and forest carbon sink. This was achieved by least-squares linear regressions[27] between historical (1961–2014) LOSS and projected grid-level NPP and grid-level HR across DGVMs:

$$NPP_{pro} \, or \, HR_{pro}(i) = a \times LOSS_{his}(i) + b \tag{1}$$

where *pro* is 'projected', *his* is 'historic', *i* is the index of model, and *a and b* are coefficients. NPP and HR were aggregated as sum within each continent and LOSS were aggregated as average across forest plot sites or within each continent (Supplementary Fig. 4). In details, we first used LOSS at forest-plot local scale to constrain the projected NPP and HR at continental scale in DGVMs and found its limited feasibility in the conventional emergent constraint approach because of spatial mismatch between data and model[25] and non-significance of EC in tropical regions (Supplementary Fig. 11). This motivated us to generate a spatially explicit map of LOSS derived from the machine learning algorithm (see Methods) to

constrain the projected NPP and HR at continental scale in DGVMs. To this end, we further used LOSS map values averaged across pixels within each continent to constrain the projected NPP and HR at continental scale in DGVMs. But we still found non-significance of EC in tropical regions (Supplementary Fig. 12), presumably because averaging LOSS across pixels still led to substantial reduction of sample size.

Alternatively, we used the historical (1961–2014) LOSS as the predictor to train a random forest model and constrain the projected (2015–2099) NPP, HR, and forest carbon sink:

$$NPP_{pro} \, or \, HR_{pro}(i) = f(LOSS_{his}(i)) \tag{2}$$

$$NPP_{pro} \, or \, HR_{pro}(i) = f(LOSS_{his}(i), MAT_{pro}, MAP_{pro}) \tag{3}$$

where Eq. 2 is the scenario without the projected climate effects and Eq. 3 is the scenario with the projected climate effects. Climate effects were incorporated to account for their potential influence on the couplings or relationships between historical LOSS and projected NPP and HR across environmental (climate) gradients. $CO_2$ was not included in Eq. (3) because of the minimal spatial heterogeneity on annual timescales. We clarify that all these variables were averages over the historical or future periods and our study thus focused on the spatial patterns of variables of interests in the quasi-steady state.

We stress that the random forest model allowed for inclusion of all grid values of variables of interests and consideration of non-linearity. This was achieved with 1000 trees and 10 maximum tree depth and 80% of the data for training purpose and the rest 20% for validation. With this approach, we surrogated the predictive relationship between the historical (1961–2014) LOSS and the projected (2015–2099) NPP and HR from a complex but physically driven DGVMs with an empirical machine learning model (random forest). Then, we feed the observed maps of LOSS derived from forest plots datasets into the trained random forest model to assess the impacts of historic LOSS on the projected NPP and HR. In this way, we were able to generate the spatially explicit[34] constraint of the projected NPP and HR, which was subsequently aggregated as sum within each continent to assess the forest carbon sink at the continental scale. Forest carbon sink was quantified by the difference of grid-level NPP and grid-level HR. The mechanistic underpinning justifying the linear (emergent) and non-linear (machine learning) constrain was the theory of faster growth and higher mortality[12,13,47] and thus couplings of growth, mortality and respiration averaged over long term and broad spatial scales[20,21].

To assess the uncertainty of our ML constraint, we conducted a bootstrapping approach to account for uncertainty from LOSS maps. As stated above, two sources of uncertainty of LOSS maps were assessed in this study, with one bootstrapping approach based on the sampled biomes of each point feature (LOSS, 100 times) and the second bootstrapping approach based on randomly sampling 90% with replacement. Here we summed up the two sources of uncertainty (standard deviation – mean × coefficient of variance) and times 2 (hereafter overall standard deviation) to account for other potential sources of uncertainty such as environmental covariates as a conservative constraint. We averaged the two ensemble means of LOSS maps from these two types of bootstrapping approaches to derive the overall average value of LOSS. Then we used a bootstrapping approach to generate 100 maps of mean value of LOSS, with its distribution following the values of the overall average and overall standard deviation of LOSS maps (see Methods). The 100 maps of mean value of LOSS were used to constrain the projected NPP or HR. The uncertainty of the constraint was assessed and ensemble means of constraints were reported.

## Data availability
Data of LOSS aggregated at 0.25 degree and the final LOSS maps (mean and standard deviation) at 0.5 degree used in our ML constraint approach are deposited in github. The raw inventory data are available upon reasonable request from the corresponding author.

## Code availability
The codes of machine learning used to generate LOSS maps were adapted from https://github.com/KailiangYu/Biogeography-of-soil-microbes.git. The code of ML constraint developed in this study is deposited in github.

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

## Acknowledgements
A.P.B. acknowledges the funding from the l'Agence Nationale de la Recherche (Make Our Planet Great Again; ANR-18-MPGA-0007). HYHC acknowledges the funding from the Natural Sciences and Engineering Research Council of Canada [RGPIN-2019-05109 and STPGP506284] and the Canadian Foundation for Innovation (36014).

## Author contributions
K.Y. designed the study with inputs from P.C. and A.P.B.; K.Y. carried out the analysis with data inputs from H.Y.H.C. and Y.H.; K.Y., P.C., and A.P.B. interpreted results; K.Y. wrote the initial manuscript draft with additional inputs from P.C., A.P.B., S.I.S., Z.L., J.B., C.D.A., H.Y.H.C., Y.H., and H.Y.

## Competing interests
The authors declare no competing interests.
