## [Peer Review File · Nature Communications]

Title: Field-based tree mortality constraint reduces estimates of model-projected forest carbon sinksREVIEWER COMMENTS

Reviewer #1 (Remarks to the Author):

The authors analyzed how integrated field-based observations of tree mortality (LOSS) alter simulated global forest dynamics. For this purpose, they investigated data from numerous forest plots to create a biomass loss map. These data were integrated into several DGVMs to constrain NPP, respiration and carbon sink projections. They have produced an excellent study by extensively bringing together field data, vegetation models and machine learning. This study is relevant because it has the potential to reduce the uncertainties in the projections of future forest carbon sinks.

It is very well shown that the simulation results partially change when integrating the observed LOSS data in global vegetation models. However, it is not clear whether this reduces really the uncertainties in NPP and HR projections (see L35). Moreover, the differences in the projections are rather small when comparing the results with and without constrains - only in South America different values for NPP and HR emerge (see Fig 4 and Table 1). Some phrases should therefore be formulated more carefully. A significance test for the differences would also be desirable here (e.g. for Fig. 4 and Table 1).

If the differences between with and without LOSS constrains are only slight, the question arises what the added value is. I would like to see a little more discussion on this point. Nevertheless, especially the biomass loss map can have a great value for the community, as it can serve as input in many further studies.

Minor issues

- It is not clear to me where I can get the biomass loss values per forest plot and the final biomass loss map. These data could be an important contribution to the community and serve as input for further studies.

- Fig. 1a: unit for biomass loss is missing

- Figure 3: Why are the drivers on the x-axis different in a-d? What is shown in c and d (a – across continents, b – tropical, c – non-tropical, d?)

- L445-L446: Makes no sense. “DGVMs refers to ... and DGVMs refers to...”

Reviewer #2 (Remarks to the Author):

This study uses ground-based measurements of forest biomass loss to mortality to reduce the enormous uncertainty surrounding forest carbon dynamics--and more specifically tree mortality-- in dynamic

global vegetation models (DGVMs). The study presents a global compilation of estimates of live biomass loss to mortality (“LOSS”), which is, to my knowledge, the largest data compilation for this variable. Comparing this to six leading DGVMs, they show that LOSS is generally overestimated. Constraining LOSS in the models, they estimate reduced forest productivity and respiration, and -- significantly-- a reduction in tropical forest C sink strength. The topic is important, and improved model improvement in the representation of LOSS is critical. The projection of lower-than-previously-estimated tropical forest C sink potential is globally significant in terms of understanding future forest feedbacks to climate change.

While the constrained models presented here appear to be a clear step forward in modeling forest dynamics, the process leaves room for significant uncertainties and biases. Probably most notably, LOSS and net primary productivity (NPP) are not fully coupled, and their decoupling drives changes in forest C stocks. I’d like to see more acknowledgement and discussion of the simplifying assumptions imbued in the model and their implications for our understanding.

I also believe that the manuscript would benefit from some work on presentation. As a forest ecologist who focuses on forest dynamics and global patterns of carbon cycling in forests but does not work with DGVMs, I (and my lab group) found it difficult to understand this paper as written. It would be improved by (1) ensuring that concepts and methods--particularly the model constraint process-- are adequately (yet concisely) explained to readers outside the modeling community, (2) clarifying some language to ensure accurate communication, and (3) ensuring that figures and tables are clearly understandable stand-alone.

Overall, I view the manuscript as important progress on a critical topic, but with large remaining uncertainties that merit further discussion and clarification.

Specific comments:

The title could be improved to something like “Constraining model-projected forest carbon sinks with field-based data reduces estimates of the global forest carbon sink”

Lines 36-38- clarify that this is decrease/ increase relative to predictions of unconstrained models

Line 38-39 - This statement seems too obvious. I’d recommend a more impactful closing to the abstract.

Lines 61-64- This doesn’t seem like a good representation even for steady state. It’s long been known that forest turnover rates largely (but don’t completely) parallel global and regional trends in productivity (e.g., Stephenson et al. 2005, Ecology Letters: <https://doi.org/10.1111/j.1461->

0248.2005.00746.x), as opposed to being a constant fraction of biomass (which doesn't vary as strongly with climate as C fluxes; see Anderson-Teixeira et al. 2021: <https://iopscience.iop.org/article/10.1088/1748-9326/abed01/meta>). But is this what's being applied in the 6 models evaluated here? Why do the patterns in Fig. 2c appear to parallel other C fluxes (e.g., NPP), as opposed to biomass (which is on average higher in the tropics but more variable in temperate regions and highest in regions like the Pacific NW)?

Line 67- a great reference on CO2 impacts on forest C cycling is Walker et al. 2021: <https://nph.onlinelibrary.wiley.com/doi/full/10.1111/nph.16866> (useful as a citation here and elsewhere).

Lines 66-68- growth also generally exceeds mortality in regrowth forests, with the ratio of the two declining as stands age. That would typically still be the case in many of the forests classified as "mature" in this study.

Line 74- "constrain" → "constraint"

Lines 75-77 - Please provide more explanation as to how EC works. The current explanation is not sufficient for readers outside of the modeling community.

Lines 100-102- It's important to note that although these variables are often coupled on broad scales, there are important instances of decoupling, which are very significant to the forest carbon balance.

Lines 28-29, 102-107, 267-272, and 1st par on p. 2 in supplement (regarding uncertainty of mortality vs ANPP records)- I do not understand/ agree with the logic that LOSS is a more certain measure than NPP when considering that minimum DBH criteria vary across censuses. It is true that a higher DBH threshold misses NPP contributions from small trees, but it also misses a similar amount of LOSS. A recently accepted paper in *New Phyt* by Piponiot et al. (should be online soon) quantified contributions of trees of different size to both variables across global forests, finding that trees <10cm contribute up to ~15% of both LOSS and NPP in mostly mature forests. Thus, there is no detectable difference in how much of a flux is missed by a high min DBH threshold. However, measurement of loss has much higher uncertainty due to the rare nature of tree mortality. Even with large plots/ long time scales, LOSS is less certain than NPP. LOSS estimates will also be affected by the decision as to whether mortality is counted on the tree or stem level. Both LOSS and NPP are subject to uncertain biomass allometries. I do not object to the use of LOSS data here, but it is incorrect to claim that it is less uncertain than NPP data.

Line 130-131- I wouldn't call the coastal Pacific northwest "warm and dry"

Line 137- The CV is also probably high here because FIA plots are small, and therefore there will be huge stochasticity in LOSS estimates. In addition, there is inherently high variability in C stocks and fluxes within the temperate zone (e.g., because of very high C stocks and fluxes in the Pacific NW).

Line 146-148- see Muller-Landau et al. 2021, *New Phyt* (<https://nph.onlinelibrary.wiley.com/doi/full/10.1111/nph.17084>)- Tansley Review on patterns of forest productivity, turnover and biomass across the tropics

Lines 159-160- Boreal forests mature more slowly than temperate forests, so why should they have a lower "maturity" threshold?

Line 173- please explain how the emergent constraint approach works

Lines 236-238- This is a big assumption that will not always hold.

Line 245-246 - "... offset projected increases in boreal and temperate forest productivity..." - this is relative to previous model predictions (as opposed to increases through time), correct? Please clarify.

Lines 244-246 - "thereby reducing the carbon sink potential of global forests." - reword to "reducing *model estimates of* the carbon sink potential of global forests"

Fig. 2- The color scale on this figure is such that intermediate values are indistinguishable from the background color. Please adjust one or the other.

Supplement p.2, par 2- Please refer (correctly) to the research networks that collected these data: e.g., RAINFOR, AfriTron, ForestGEO (I believe this is what's being called "STRI", although it's not straightforward to trace). It would also be appropriate to acknowledge their data contributions in the main text.

An additional source of data would be the ForC database: <https://forc-db.github.io/>

P. 3 of supplement - "we derived aboveground biomass loss from mortality ($\text{kg m}^{-2} \text{yr}^{-1}$) - what is this supposed to mean? They are the same thing, right?"

Geospatial modeling and environmental drivers (described in SI) - I suspect that the approach described here could very easily be resulting in over-fitting. An R^2 of 0.93 is suspicious. The map produced (Fig. 2a) looks fairly reasonable, at least on a macroscopic scale, so it's unlikely that this affects the study's conclusions.

Supplementary Fig. 3- please provide a legend explaining the symbol colors

Supplementary Fig. 13- Please explain "RF surrogate model"

Throughout- this isn't critical, but at least in the empirical literature, C fluxes are more commonly reported in Mg C/ha/yr or $\text{g C/m}^2/\text{yr}$, as opposed to $\text{kg C/m}^2/\text{yr}$. Also, kg should not be capitalized.

Reviewer #1 (Remarks to the Author):

The authors analyzed how integrated field-based observations of tree mortality (LOSS) alter simulated global forest dynamics. For this purpose, they investigated data from numerous forest plots to create a biomass loss map. These data were integrated into several DGVMs to constrain NPP, respiration and carbon sink projections. They have produced an excellent study by extensively bringing together field data, vegetation models and machine learning. This study is relevant because it has the potential to reduce the uncertainties in the projections of future forest carbon sinks.

Response: We appreciate the reviewer's positive comments on this study. We carefully addressed the reviewer's suggestions and comments and provided two versions of revised manuscript – tracked change and clean version with changes being highlighted as red color. In the following responses to reviewers, line numbers were from the clean version of the revised manuscript.

It is very well shown that the simulation results partially change when integrating the observed LOSS data in global vegetation models. However, it is not clear whether this reduces really the uncertainties in NPP and HR projections (see L35). Moreover, the differences in the projections are rather small when comparing the results with and without constrains - only in South America different values for NPP and HR emerge (see Fig 4 and Table 1). Some phrases should therefore be formulated more carefully. A significance test for the differences would also be desirable here (e.g. for Fig. 4 and Table 1).

Response: Thank you for the recommendation. In the revised manuscript, we used ANOVA test to explicitly evaluate whether the difference is significant between before and after constraint of projected NPP and HR. Please see the results of Fig. 4, Table 1 and Appendix Fig. 19 for details. The results show that the constraint by the machine learning approach significantly reduced the model spread in grid-level NPP_{pro} and HR_{pro} generally in tropical regions and particularly in South America (Fig. 4; Table 1) (see lines 244-246 in the clean version of revised manuscript).

If the differences between with and without LOSS constrains are only slight, the question arises what the added value is. I would like to see a little more discussion on this point. Nevertheless, especially the biomass loss map can have a great value for the community, as it can serve as input in many further studies.

Response: We found that the difference between before and after the constraint of the projected carbon sink was not significant in North America. We hypothesized that this pattern was partially caused by the general overestimation of LOSS in models in eastern North America – mostly temperate forests and general underestimation of LOSS in models in other regions – mostly boreal forests. Thus, there was compensation or trade-off when the constrain was implemented at the whole North America scale. To examine this hypothesis, we further divided the North America into temperate and boreal forests. The results confirmed the significant difference of projected NPP and HR (see Supplementary Fig. 17 for details) in temperate and boreal forests, respectively, between with and without LOSS constraint. These

results highlight the importance of spatial scale in ML constraint approach. We thus recommend accounting for the role of spatial trade-off in our ML constraint approach or using our ML constraint approach at broad spatial scales whereby the effect of spatial trade-off is minimal. These points have been clarified in the revised manuscript (lines 247-255).

To further clarify the value of our machine learning approach, we stated 'This ML approach thus allowed for inclusion of all gridded patterns of LOSS and a spatially explicit constraint of projected NPP and HR in each model which can be sequentially aggregated at broad spatial scales, while we note that the couplings of LOSS, NPP and HR could be weak at local or pixel scales.' (lines 132-135) and 'Our results show the potential of leveraging a machine-learning (ML) approach of constraint and forest demography data to constrain the projected NPP and HR at broad spatial scales. This ML approach accounts for the non-linearity and all pixel values of variables of interest (i.e., LOSS here), which is not necessarily considered using the conventional EC approach.' (lines 286-290).

We appreciate the reviewer's positive comment on the value of LOSS map and will make the LOSS map publicly available in scientific communities.

Minor issues

- It is not clear to me where I can get the biomass loss values per forest plot and the final biomass loss map. These data could be an important contribution to the community and serve as input for further studies.

Response: Here we submitted the raw data of biomass loss from mortality aggregated at 0.25 degree and the final LOSS maps (mean and standard deviation) at 0.5 degree used in our ML constraint approach. Once the paper is accepted, we will deposit these data products in github.

Moreover, here we gave more information for the core codes used in this study. The codes of machine learning used to generate LOSS maps were adapted from <https://github.com/KailiangYu/Biogeography-of-soil-microbes.git>. The code of ML constraint developed in this study was submitted for review and will be deposited in github once the paper is accepted.

The data and code availability are also provided in the revised manuscript.

- Fig. 1a: unit for biomass loss is missing

Response: Done.

- Figure 3: Why are the drivers on the x-axis different in a-d? What is shown in c and d (a – across continents, b – tropical, c – non-tropical, d?)

Response: We first identified mean annual temperature (MAT), aridity index (AI), and precipitation seasonality as the dominant predictors of LOSS across continents (Supplementary Fig.7a). These factors were found to be not important in tropical regions, whereby the soil property gradient instead of climate gradient was high.

Thus, in tropical regions, we tested the effects of soil property on LOSS instead of climate and the results showed the importance of soil property such as cation exchange capacity and soil nitrogen.

In terms of figure 3c and figure 3d, two different metrics of the competition were used. Basal area is strongly correlated with total biomass, and higher LOSS in the higher basal area is attributed to the correlation. This likely impairs the conclusion of the role of competition in LOSS. Thus, instead, we also used another competition metrics – SDI to further confirm the role of competition in LOSS. These points have been clarified in the legend of figure 3 in the revised manuscript (lines 494-497).

- L445-L446: Makes no sense. "DGVMs refers to ... and DGVMs refers to..."

Response: Done.

Reviewer #2 (Remarks to the Author):

This study uses ground-based measurements of forest biomass loss to mortality to reduce the enormous uncertainty surrounding forest carbon dynamics--and more specifically tree mortality-- in dynamic global vegetation models (DGVMs). The study presents a global compilation of estimates of live biomass loss to mortality ("LOSS"), which is, to my knowledge, the largest data compilation for this variable. Comparing this to six leading DGVMs, they show that LOSS is generally overestimated. Constraining LOSS in the models, they estimate reduced forest productivity and respiration, and -- significantly-- a reduction in tropical forest C sink strength. The topic is important, and improved model improvement in the representation of LOSS is critical. The projection of lower-than-previously-estimated tropical forest C sink potential is globally significant in terms of understanding future forest feedbacks to climate change.

Response: We appreciate the reviewer's positive comments in terms of data coverage, topic, implication and significance of conclusion in this study. We carefully addressed the reviewer's suggestions and comments and provided two versions of revised manuscript – tracked change and clean version with changes being highlighted as red color. In the following responses to reviewers, line numbers were from the clean version of the revised manuscript.

While the constrained models presented here appear to be a clear step forward in modeling forest dynamics, the process leaves room for significant uncertainties and biases. Probably most notably, LOSS and net primary productivity (NPP) are not fully coupled, and their decoupling drives changes in forest C stocks. I'd like to see more acknowledgement and discussion of the simplifying assumptions imbued in the model and their implications for our understanding.

Response: We agreed with the reviewer and recognized the potential uncertainty of our approach. Thus, in the revised manuscript, we used a bootstrapping approach (100 times) instead of the original constraint approach using one LOSS map of the main text to assess the uncertainty of our ML constraint. In details, in the methods section, we clarified "To assess the uncertainty of our ML constraint, we conducted a bootstrapping approach to account for uncertainty from LOSS maps. As stated

above, two sources of uncertainty of LOSS maps were assessed in this study, with one bootstrapping approach based on the sampled biomes of each point feature (LOSS, 100 times) and the second bootstrapping approach based on randomly sampling 90% with replacement. Here we summed up the two sources of uncertainty (standard deviation – mean \times coefficient of variance) and times 2 (hereafter overall standard deviation) to account for other potential sources of uncertainty such as environmental covariates as a conservative constraint. We averaged the two ensemble means of LOSS maps from these two types of bootstrapping approaches to derive the overall average value of LOSS. Then we used a bootstrapping approach to generate 100 maps of mean value of LOSS, with its distribution following the values of the overall average and overall standard deviation of LOSS maps (see Methods). The 100 maps of mean value of LOSS were used to constrain the projected NPP or HR. The uncertainty of the constraint was assessed and ensemble means of constraints were reported.”. (lines 413-426)

In the section of results and discussion, we clarified, ‘Ultimately, we used a bootstrap approach to generate 100 maps of mean value of LOSS with its distribution following the values of the average and 2 times of standard deviation of LOSS maps as a conservative constraint (see Methods). Then the 100 maps of mean value of LOSS were used to constrain the projected NPP or HR as ensemble means in our ML constraint and the uncertainty of the constraint was assessed’ (lines 222-226)

In terms of the couplings of LOSS, NPP, and HR, our original manuscript stated that this coupling was largely valid for broad spatial scales, while the instantaneous rates of NPP, LOSS and HR could be decoupled over a short term or local scales. In the revised manuscript, we further clarify that these couplings could change with time due to the potential effects of changes in usage efficiency of resources (water and nutrients) under global change. Another non-modeled factor of decoupling is the role of climate-induced disturbances that can strongly increase LOSS and have a delayed positive or negative effect on NPP in the recovery phase. Furthermore, our approach allows for an explicitly spatial constrain of projected NPP and HR at large scale, but the couplings of LOSS, NPP, and HR could be weak at local or pixel scale. These points have been more clearly clarified in the revised manuscripts (lines 72-75, 276-281).

Our results also showed that LOSS constraint had a greater impact on reducing NPP than HR, resulting in a net NEE reduction. This may tell us something about the strength of coupling between LOSS and productivity vs HR (decomposition). Thus, in the revised manuscript, we clarified ‘HR remains even more poorly understood and more simply represented than LOSS and NPP in DGVMs. Thus, the couplings of LOSS with HR could be weaker or more uncertain (subject to changes) relative to the couplings of LOSS and NPP at long term and broad spatial scales. This presumably explains why our ML constraint has a greater influence on NPP than HR found in this study’ (lines 282-286).

I also believe that the manuscript would benefit from some work on presentation. As a forest ecologist who focuses on forest dynamics and global patterns of carbon cycling in forests but does not work with DGVMs, I (and my lab group) found it difficult to understand this paper as written. It would be improved by (1) ensuring that concepts and methods--particularly the model constraint process-- are adequately

(yet concisely) explained to readers outside the modeling community, (2) clarifying some language to ensure accurate communication, and (3) ensuring that figures and tables are clearly understandable stand-alone.

Response: As recommended, we have improved the presentations in these three aspects. In particular, we now clarify the emergent constraint so that readers outside the modeling community could better understand it. For instance, we state, 'The essence of such an EC approach is to examine the statistical relationships between historical and projected variables of interest in a multi-model ensemble, whereby the historical observations are used to reduce the uncertainty of model projections²⁵. This empirical EC approach is complementary to the bottom-up approach in which data or process optimization (sometimes through data assimilation) is applied to improve model projections²⁶'. (lines 80-85).

Furthermore, we fundamentally restated the legend of supplementary Fig. 4 in which the schematic illustration of two approaches to constrain the projected forest carbon sink across continents was more clearly clarified.

Lastly, the legends of figures in both the main text and appendix were checked again so that they could be clearly understandable stand-alone.

Overall, I view the manuscript as important progress on a critical topic, but with large remaining uncertainties that merit further discussion and clarification.

Response: We appreciate the reviewer's positive comments. As responded above, we used a bootstrapping approach (100 times) to assess the uncertainty of our constrain of projected NPP and HR and the results show low uncertainty of our ML constraint. The points about uncouplings, presentation in terms of the constraint have been addressed following the reviewer's suggestion.

Specific comments:

The title could be improved to something like "Constraining model-projected forest carbon cycling with field-based data reduces estimates of the global forest carbon sink"

Response: Done.

Alternatively, here we provided another similar title for potential choice:

'Field-based tree mortality constraint reduces estimates of model-projected forest carbon sinks'

Lines 36-38- clarify that this is decrease/ increase relative to predictions of unconstrained models

Response: Done.

Line 38-39 - This statement seems too obvious. I'd recommend a more impactful closing to the abstract.

Response: We have rephased it as 'These results suggest the feasibility of using forest demographic data to empirically constrain forest carbon sink projections, including reductions in projected tropical forest carbon sinks.'

Lines 61-64- This doesn't seem like a good representation even for steady state. It's long been known that forest turnover rates largely (but don't completely) parallel global and regional trends in productivity (e.g., Stephenson et al. 2005, Ecology Letters: <https://doi.org/10.1111/j.1461-0248.2005.00746.x>), as opposed to being a constant fraction of biomass (which doesn't vary as strongly with climate as C fluxes; see Anderson-Teixeira et al. 2021: <https://iopscience.iop.org/article/10.1088/1748-9326/abed01/meta>). But is this what's being applied in the 6 models evaluated here? Why do the patterns in Fig. 2c appear to parallel other C fluxes (e.g., NPP), as opposed to biomass (which is on average higher in the tropics but more variable in temperate regions and highest in regions like the Pacific NW)?

Response: In the revised manuscript, we have rephrased as 'While the fraction of biomass subjected to tree mortality is often treated as a proportion (constant or varying) of standing stocks in ESM simulations¹³, this simple model representation of mortality and growth is commonly not observed in forests responding to global change.'

Line 67- a great reference on CO2 impacts on forest C cycling is Walker et al. 2021: <https://nph.onlinelibrary.wiley.com/doi/full/10.1111/nph.16866> (useful as a citation here and elsewhere).

Response: As recommended, we added this reference in the revised manuscript.

Lines 66-68- growth also generally exceeds mortality in regrowth forests, with the ratio of the two declining as stands age. That would typically still be the case in many of the forests classified as "mature" in this study.

Response: in the revised manuscript, we added this aspect about regrowth forests. 'In contrast, growth may be exceeding mortality in regrowing secondary forests¹⁶ or due to...'

Line 74- "constrain" → "constraint"

Response: Done

Lines 75-77 - Please provide more explanation as to how EC works. The current explanation is not sufficient for readers outside of the modeling community.

Response: In the revised manuscript, as responded above, we further clarified, 'The essence of such an EC approach is to examine the statistical relationships between historical and projected variables of interest in a multi-model ensemble, whereby the historical observations are used to reduce the uncertainty of model projections²⁵. This empirical EC approach is complementary to the bottom-up approach in which data or process optimization (sometimes through data assimilation) is applied to

improve model projections²⁶. (lines 80-85). Moreover, we fundamentally improved the legend of supplementary Fig. 4 so that the readers outside of modeling community could better understand the EC and our ML constraint approach.

Lines 100-102- It's important to note that although these variables are often coupled on broad scales, there are important instances of decoupling, which are very significant to the forest carbon balance.

Response: As responded above, we have revised it to 'these couplings could be altered with changed usage efficiency of resources (i.e., water and nutrients) under global change'. 'Another non-modeled factor of decoupling is the role of climate-induced disturbances that could strongly increase LOSS and have a delayed positive or negative effect on NPP in the recovery phase'. We also further clarified 'This ML approach thus allowed for the inclusion of all grid variables of LOSS and a spatially explicit constraint of projected NPP and HR in each model which can be sequentially aggregated at broad spatial scales, while we note that the couplings of LOSS, NPP and HR could be weak at local or pixel scales'.

Lines 28-29, 102-107, 267-272, and 1st par on p. 2 in supplement (regarding uncertainty of mortality vs ANPP records)- I do not understand/ agree with the logic that LOSS is a more certain measure than NPP when considering that minimum DBH criteria vary across censuses. It is true that a higher DBH threshold misses NPP contributions from small trees, but it also misses a similar amount of LOSS. A recently accepted paper in New Phyt by Piponiot et al. (should be online soon) quantified contributions of trees of different size to both variables across global forests, finding that trees <10cm contribute up to ~15% of both LOSS and NPP in mostly mature forests. Thus, there is no detectable difference in how much of a flux is missed by a high min DBH threshold. However, measurement of loss has much higher uncertainty due to the rare nature of tree mortality. Even with large plots/ long time scales, LOSS is less certain than NPP. LOSS estimates will also be affected by the decision as to whether mortality is counted on the tree or stem level. Both LOSS and NPP are subject to uncertain biomass allometries. I do not object to the use of LOSS data here, but it is incorrect to claim that it is less uncertain than NPP data.

Response: In the revised manuscript, we have rephrased as 'We used LOSS also because: 1) it can be directly measured in forest inventory datasets¹³ with high accuracy; 2) LOSS remains less studied relative to NPP and LOSS is unrealistically represented (i.e., as a proportion of NPP) in DGVMs¹³. Thus, more work with LOSS data products is urgently needed in scientific communities' (lines 111-115).

Moreover, LOSS rather than NPP is more mechanistically linked with HR based on the first principles, thus allowing us to use LOSS to constrain both projected NPP and HR. We thus clarified '3) LOSS rather than NPP is more mechanistically linked with HR based on first principles involving processes of decomposition following LOSS²³'.

Line 130-131- I wouldn't call the coastal Pacific northwest "warm and dry"

Response: Done.

Line 137- The CV is also probably high here because FIA plots are small, and therefore there will be huge stochasticity in LOSS estimates. In addition, there is inherently high variability in C stocks and fluxes within the temperate zone (e.g., because of very high C stocks and fluxes in the Pacific NW).

Response: Thanks for these suggestions. In the revised manuscript, we followed the reviewer's suggestion and clarified these points. " (see line 152)

Line 146-148- see Muller-Landau et al. 2021, New Phyt (<https://nph.onlinelibrary.wiley.com/doi/full/10.1111/nph.17084>)- Tansley Review on patterns of forest productivity, turnover and biomass across the tropics

Response: We added this reference in the revised manuscript.

Lines 159-160- Boreal forests mature more slowly than temperate forests, so why should they have a lower "maturity" threshold?

Response: Compared with temperate forest tree species, boreal tree species have shorter longevities. Also trees in boreal forests grow slowly and thus could mature (in terms of relatively steady growth, biomass from mortality and turnover) at a relatively younger age than temperate forests.

Line 173- please explain how the emergent constraint approach works

Response: In the revised manuscript, we further clarified 'This approach was conducted by building the statistic (linear) relationship of the historical LOSS averaged at forest-plot scale (derived from original plot data of LOSS) or continental scale (derived from the map of LOSS) and projected NPP and HR aggregated as sums across continents (see Methods and Supplementary Fig. 4 for details)'.

Lines 236-238- This is a big assumption that will not always hold.

Response: in the revised manuscript, we rephrased as 'Our approach assumes that this coupling holds both in the short-term historical period and will hold in the future, thus allowing us...'.

Line 245-246 - "... offset projected increases in boreal and temperate forest productivity..." - this is relative to previous model predictions (as opposed to increases through time), correct? Please clarify.

Response: In the revised manuscript, we clarified this point as suggested by the reviewer. 'Our results indicate that the projected increase in tropical forest productivity after constraint may not be as great as previously thought relative to predictions of unconstrained models' (line 290-292)

Lines 244-246 - "thereby reducing the carbon sink potential of global forests." - reword to "reducing *model estimates of* the carbon sink potential of global forests"

Response: Done.

Fig. 2- The color scale on this figure is such that intermediate values are indistinguishable from the background color. Please adjust one or the other.

Supplement p.2, par 2- Please refer (correctly) to the research networks that collected these data: e.g., RAINFOR, AfriTron, ForestGEO (I believe this is what's being called "STRI", although it's not straightforward to trace). It would also be appropriate to acknowledge their data contributions in the main text.

An additional source of data would be the ForC database: <https://forc-db.github.io/>

Response: We tried to follow the reviewer's suggestion for the color scale. Later we found that the current color scale is useful to differentiate the regions of overestimation of LOSS vs underestimation of LOSS in DGVMs before and after constraint.

In the revised manuscript, we have added these data networks. 'The mature forest plot datasets in tropical regions were mainly derived from three pieces of literature with original data sources from RAINFOR, AfriTron, and ForestGEO⁸⁻¹⁰' We also clarified the ForC database as an additional source of data 'The readers could also refer to other potential sources of data such as Global Forest Carbon Database (ForC)¹¹ <https://forc-db.github.io/> to augment the database used in this study.'

P. 3 of supplement - "we derived aboveground biomass loss from mortality (kg m⁻² yr⁻¹) - what is this supposed to mean? They are the same thing, right?"

Response: we now changed it to 'In our study, we used the available data of aboveground biomass loss from mortality...'

Geospatial modeling and environmental drivers (described in SI) - I suspect that the approach described here could very easily be resulting in over-fitting. An R² of 0.93 is suspicious. The map produced (Fig. 2a) looks fairly reasonable, at least on a macroscopic scale, so it's unlikely that this affects the study's conclusions.

Response: Geospatial modeling through machine learning (random forest) approach generally have better R² of the results or patterns with consideration of non-linearity et al, relative to the case of using linear regressions to extrapolate at a global scale. Furthermore, the R² also depends on the type of the variable used and our test shows that R² for LOSS was higher than the turnover rate, which is derived from biomass divided by LOSS. The relatively high R² in this study is also partially because of the focus of mature forests with the potential low influence of forest age or other random effects.

Supplementary Fig. 3- please provide a legend explaining the symbol colors

Response: In the revised manuscript, we used black color instead of symbol colors. The symbol colors were achieved in the original R package. We found that this R package is now uninstalled and thus currently not available to use any more.

Supplementary Fig. 13- Please explain “RF surrogate model”

Response: Done.

Throughout- this isn't critical, but at least in the empirical literature, C fluxes are more commonly reported in Mg C/ha/yr or g C/m²/yr, as opposed to kg C/m²/yr. Also, kg should not be capitalized.

Response: In the revised manuscript, we changed all of the units of kg C/m²/yr to Mg C/ha/yr.

REVIEWERS' COMMENTS

Reviewer #1 (Remarks to the Author):

Many thanks to the authors for the very good revision of their manuscript. In the answers, they addressed all my concerns very detailed. Very valuable are the statistical tests for significance for the differences between without and with constraints. I also like the distinction between temperate and boreal forest in North America (see Fig S17). I think this result is very important and I would recommend that this distinction should also be shown in Fig 4. I also very much appreciate the authors' commitment to make all methods and results (especially the LOSS map) available online.

I still have challenges in understanding with Figure 3, the meaning of 3c and 3d are difficult to explore. But I think I have understood it now. It is the same analysis of the non-tropical regions with a linear mixed model with (c) basal area as competition index and (d) SDI as competition index. If I have understood this correctly, please adjust the figure caption accordingly.

Reviewer #2 (Remarks to the Author):

I thank the authors or responses to my previous concerns. I find the manuscript to be significantly improved.

Specific comments:

Title- Thank you for implementing my suggestion for a title change. I personally prefer the alternative in the response to reviews ('Field-based tree mortality constraint reduces estimates of model-projected forest carbon sinks'), but feel strongly that the authors should go with their preference.

Line 39- Wording of this sentence is awkward. I'd suggest something like "..., suggesting that projected tropical forest carbon sinks may be overestimated."

Line 61- ESMs don't demonstrate; rather, they attempt to replicate reality. Please reword to something like, "... and is consistent with ESM projections across forest biomes"

Line 74- these are also likely to be changed by altered resource availability

Lines 76, 112- certainty depends on plot size (and other factors)—small plots (e.g., FIA) will yield data with high uncertainty. Accuracy also depends on the biomass allometry equations used. I don't find the

“with high accuracy” claim appropriate.

Lines 115-117- I’m not convinced that this (#3) is true. Woody productivity, and hence LOSS, make up only a fairly small fraction of GPP (~1/6th), and a lot of other components of GPP would be linked to HR on much shorter time scales (e.g., fine root productivity, foliage production/litterfall).

Lines 125-130- Does this mean simply replacing the modeled value with one “corrected” based on the linear relationship? Does this happen for each modeled year, and then carried forward, or are all values corrected at the end? (I feel that more explanation is still required for readers who are not familiar with the technique.)

Lines 77-79 – It seems more appropriate to say “... it remains unclear *how* ground-based datasets...” (as opposed to *whether*)

Line 327 – please clarify what is meant by tree-level and non-tree level

Line 397- clarified ◊ clarify

Line 429- Nothing is mentioned about data availability for the raw data compilation. This would be valuable to the research community. Please make that available, along with sources from which it was obtained.

Fig. 2- I’d recommend making a grey background on the continents so that values coded white can be distinguished from no data.

SI line 46 – “STRI plots” here should be referred to as ForestGEO (assuming they are ForestGEO, which they most likely are)

REVIEWERS' COMMENTS

Reviewer #1 (Remarks to the Author):

Many thanks to the authors for the very good revision of their manuscript. In the answers, they addressed all my concerns very detailed. Very valuable are the statistical tests for significance for the differences between without and with constraints. I also like the distinction between temperate and boreal forest in North America (see Fig S17). I think this result is very important and I would recommend that this distinction should also be shown in Fig 4. I also very much appreciate the authors' commitment to make all methods and results (especially the LOSS map) available online.

Response: *We appreciate the positive comments of the reviewer on our revised manuscript. To follow the reviewer's suggestion, we added in the legend of Fig. 4 'The constraint effect was significant when North America were divided into temperate and boreal forests (see results of Supplementary Fig. S17).'*

I still have challenges in understanding with Figure 3, the meaning of 3c and 3d are difficult to explore. But I think I have understood it now. It is the same analysis of the non-tropical regions with a linear mixed model with (c) basal area as competition index and (d) SDI as competition index. If I have understood this correctly, please adjust the figure caption accordingly.

Response: *we thank the reviewer's suggestion. In the revised manuscript, we rewrote the legend of figure 3 and clarified the basal area and SDI as competition indices. 'Panels c and d used basal area and stand density index (SDI) as competition index, respectively.'*

Reviewer #2 (Remarks to the Author):

I thank the authors or responses to my previous concerns. I find the manuscript to be significantly improved.

Response: *We appreciate the positive comments of the reviewer on our revised manuscript.*

Specific comments:

Title- Thank you for implementing my suggestion for a title change. I personally prefer the alternative in the response to reviews ('Field-based tree mortality constraint reduces estimates of model-projected forest carbon sinks'), but feel strongly that the authors should go with their preference.

Response: Thanks for the reviewer. We also feel that the title ‘Field-based tree mortality constraint reduces estimates of model-projected forest carbon sink’ is better. Thus, we used this title in the revised manuscript.

Line 39- Wording of this sentence is awkward. I’d suggest something like “..., suggesting that projected tropical forest carbon sinks may be overestimated.”

Response: We restated as ‘These results highlight the feasibility of using forest demographic data to empirically constrain forest carbon sink projections and the potential overestimation of projected tropical forest carbon sinks’.

Line 61- ESMs don’t demonstrate; rather, they attempt to replicate reality. Please reword to something like, “... and is consistent with ESM projections across forest biomes”

Response: Done

Line 74- these are also likely to be changed by altered resource availability

Response: Done

Lines 76, 112- certainty depends on plot size (and other factors)—small plots (e.g., FIA) will yield data with high uncertainty. Accuracy also depends on the biomass allometry equations used. I don’t find the “with high accuracy” claim appropriate.

Response: In the revised manuscript, we deleted the statement of with high uncertainty in line 76. In line 112, we used ‘with high uncertainty relative to remote sensing’.

Lines 115-117- I’m not convinced that this (#3) is true. Woody productivity, and hence LOSS, make up only fairly small fraction of GPP (~1/6th), and a lot of other components of GPP would be linked to HR on much shorter time scales (e.g., fine root productivity, foliage production/litterfall).

Response: In the revised manuscript, we deleted point 3.

Lines 125-130- Does this mean simply replacing the modeled value with one “corrected” based on the linear relationship? Does this happen for each modeled year, and then carried forward, or are all values corrected at the end? (I feel that more explanation is still required for readers who are not familiar with the technique.)

Response: we thank the reviewer for pointing this out. In the revised manuscript, we added one more sentence to clarify this. ‘In this sense, each model was treated as a sample to fit the heuristic (linear) relationship between LOSS and projected NPP and HR and the observational LOSS (mean \pm sd) was then used to impose the constraint on projected NPP and HR (see Supplementary Fig. 4 for more details).’

In the legends of Supplementary Fig. 4, we also further clarified ‘See examples of the supplementary Fig. 11 and Fig. 12 for details.’ To this end, the readers would more clearly understand the conventional constraint approach.

Lines 77-79 – It seems more appropriate to say “... it remains unclear **how** ground-based datasets...” (as opposed to **whether**)

Response: *Nice catch.*

Line 327 – please clarify what is meant by tree-level and non-tree level

Response: *we restated as ‘decomposed into components of HR contributed by trees vs non-trees’*

Line 397- clarified \diamond clarify

Response: *Done*

Line 429- Nothing is mentioned about data availability for the raw data compilation. This would be valuable to the research community. Please make that available, along with sources from which it was obtained.

Response: *Our manuscript clarified the details of compiling our plot datasets in section ‘Forest plots’ and supplementary materials. Here in the revised manuscript, we further clarified ‘The raw inventory data are available upon reasonable request from the corresponding author.’*

Fig. 2- I’d recommend making a grey background on the continents so that values coded white can be distinguished from no data.

Response: *We appreciate the reviewer’s suggestion and tried to change Fig. 2 and other figures to grey background. But we found that the grey background has not be widely used in figures. Alternatively, we could provide an appendix to show where our data has been extrapolated, if needed.*

SI line 46 – “STRI plots” here should be referred to as ForestGEO (assuming they are ForestGEO, which they most likely are)

Response: *Nice catch. Done.*